# Effect of 6-Week Instrument-Assisted Soft Tissue Mobilization on Joint Flexibility and Musculotendinous Properties

**DOI:** 10.3390/sports12060150

**Published:** 2024-05-28

**Authors:** Naoki Ikeda, Kazuya Hiratsuka, Tadao Isaka

**Affiliations:** 1Institute for General Education, Ritsumeikan University, 1-1-1 Noji-Higashi, Kusatsu 525-8577, Shiga, Japan; hrtk@hachinohe-u.ac.jp; 2Faculty of Sport and Health Science, Ritsumeikan University, 1-1-1 Noji-Higashi, Kusatsu 525-8577, Shiga, Japan; isaka@se.ritsumei.ac.jp

**Keywords:** muscle and tendon elasticity, stiffness, range of motion, stretch tolerance, shear wave elastography, training effect

## Abstract

Instrument-assisted soft tissue mobilization (IASTM) stimulates soft subcutaneous tissues by applying pressure to the skin with a specialized bar or spurtle-like instrument. No studies have verified whether several weeks of continuous IASTM alone can alter joint flexibility and musculotendinous properties in healthy participants. We examined the effect of a 6-week IASTM program on joint flexibility and the musculotendinous properties of the lower limbs. Fourteen healthy men (aged 19–35 years) who participated in a 6-week IASTM program (3 days weekly) for the soft tissue of the posterior aspect of one lower leg were included. The other leg served as the control. Before and after the intervention, we measured the maximal ankle joint dorsiflexion angle (dorsiflexion range of motion: DFROM) and maximal passive torque (MPT), a measure of stretch tolerance. We measured muscle and tendon stiffness using shear wave elastography on the gastrocnemius and Achilles tendon. IASTM significantly increased the DFROM and MPT (*p* < 0.05 for both). However, no significant changes were observed in muscle and tendon stiffness. None of the parameters changed significantly in the control group. The 6-week IASTM program increased stretch tolerance and joint flexibility but did not change muscle and tendon stiffness.

## 1. Introduction

Instrument-assisted soft tissue mobilization (IASTM) is a procedure in which mechanical stimuli, such as compression and shear stress, are repeatedly applied to soft subcutaneous tissues (muscles, overlying deep fascia, and tendons) at various intensities by stroking the skin with a bar or spurtle-like instrument [1,2,3]. In rehabilitation settings, IASTM resolves scar tissue, adhesions, and restricted fascial mobility [1,3]. This technique has also been adopted in sports. It is increasingly used by athletes before and after practice or games to regulate physical conditions such as joint flexibility, range of motion (ROM), and subjective muscle stiffness. Although there is no consensus [4], some previous studies examining the acute effects of IASTM have reported that joint flexibility improves immediately after the procedure [5,6,7]. However, the effects of several weeks of continuous IASTM on joint flexibility and the musculotendinous properties of target muscles remain unclear.

In clinical settings, IASTM is often combined with other treatment modalities, such as resistance exercise or stretching, and previous studies have reported the effect of several weeks of continuous exercise involving IASTM combined with other treatment modalities on joint flexibility [8,9]. For example, a previous study reported that 6 weeks of mixed treatment modalities (IASTM, resistance exercise, and stretching) in healthy volunteers were more effective in improving joint flexibility than stretching alone [8]. Additionally, another previous study reported that 3 weeks of mixed treatment modalities (IASTM, resistance exercise, and stretching) in physically active volunteers was more effective in improving joint flexibility than no treatment [9]. The former study did not implement IASTM-only conditions, unlike the latter study; however, no statistically significant differences were reported between IASTM-only and mixed treatment modalities, as well as between IASTM-only and no treatment. To our knowledge, the effects of IASTM alone have not been established. When instructors prescribe treatment modalities, they consider the effectiveness of each treatment modality. Therefore, clarifying the efficacy of IASTM alone as a treatment modality for improving joint flexibility is vital.

Notably, many factors are associated with joint flexibility, and some of the most influential factors are the mechanical properties, such as the stiffness of the muscle–tendon unit (MTU) that spans the joint and its connective tissue (joint capsule and ligaments) [10]. Joint flexibility improves with decreased muscle and tendon stiffness, increasing the MTU’s extensibility [11,12]. IASTM is thought to increase the extensibility of target soft tissues [13]. Moreover, a previous study has suggested that the stiffness of the target MTU is reduced immediately after IASTM [7]. Therefore, joint flexibility is expected to improve following the reduction of musculotendinous stiffness after several weeks of continuous IASTM alone. However, changes in neural properties, such as the stretch reflex, pain sensation, and maximum stretch tolerance during MTU extension, are also influential factors in improving joint ROM [10,11,12]. IASTM is thought to inhibit pain receptor responses [13]; a previous study has reported pain reduction after a 4-week IASTM program, which included resistance exercise for patients with chronic low back pain [14]. Therefore, mechanical stimulation of the skin through IASTM could inhibit the activity of the subcutaneous target muscles by altering the response of proprioceptors, such as muscle spindles, or modulating the response of the central nervous system. However, no studies have verified whether several weeks of continuous IASTM alone can alter stretch tolerance in healthy participants. 

This study aimed to investigate the effects of a 6-week continuous self-implemented IASTM (self-IASTM) program for the plantar flexor muscles and Achilles tendon (AT) on the mechanical and neural properties of the triceps surae. We hypothesized that IASTM would decrease musculotendinous stiffness and change stretch tolerance to improve joint flexibility.

## 2. Materials and Methods

### 2.1. Participants

This study included 14 healthy men (age, 21 ± 5 years; height, 1.73 ± 0.06 m; body weight, 67.8 ± 19.1 kg; mean ± standard deviation). They had no orthopedic diseases in the lower extremities (muscle, tendon, ligament injuries, and peripheral neuropathy). They had participated in sports activities (ball games such as softball, tennis, and badminton [*n* = 9], cycling [*n* = 2], resistance exercise [*n* = 2], running [*n* = 1]) at the recreational level (once or twice per week [*n* = 9], 3 or 4 times per week [*n* = 5]). We also screened participants to ensure they did not have restricted posterior talar glide and bony dorsiflexion ROM (DFROM) anteriorly at the talocrural joint. We performed a posterior talar glide inhibition test involving the flexor hallucis longus tendon to screen for these factors and preliminarily determine whether bone impingement pain occurred at the anterior joint cleft of the talocrural joint during ankle dorsiflexion. We instructed the participants not to start any new flexibility training besides the intervention during the intervention period, and they provided a signed consent form after we explained the content, purpose, benefits, and risks associated with their participation in the study. This study was conducted after review and approval by our institution’s Ethical Review Committee for Research Involving Human Subjects (approval number BKC-LSMH-2021-055).

### 2.2. Study Design

This study was designed to test the effect of the 6-week IASTM program on joint flexibility and the musculotendinous properties of the lower limbs. Fourteen healthy men participated in this study. Both legs were measured in all participants; one leg underwent the 6-week IASTM program on the posterior aspect of the lower leg, and the other did not undergo IASTM (control leg). All measurements were obtained in the same order and for both legs. The leg that underwent IASTM, regardless of being the dominant or non-dominant leg, was randomized. The following dependent variables were evaluated before and after the intervention: DFROM, maximal passive torque (a measure of stretch tolerance), and muscle and tendon stiffness. 

### 2.3. IASTM

The intervention involving IASTM included soft tissue mobilization techniques using instruments (ScandSlick, Faslic Inc., Tokyo, Japan) and the fascia slick technique (Figure 1a), and it was performed according to the protocol developed by a therapist who completed training in IASTM (Graston Technique^®^, Indianapolis, IN, USA) Modules 1 (basic) and 2 (advanced) and had over 10 years of experience performing IASTM. This protocol was based on a previous study showing that joint flexibility improves immediately after 5 min of IASTM [7]. The IASTM participants performed repeated mechanical stimulation of the posterior leg’s soft tissue structures (skeletal muscles, such as the medial [MG] and lateral [LG] heads of the gastrocnemius; soleus [SOL]; and tibialis posterior, flexor hallucis longus, and flexor digitorum longus, including the deep fascia overlying those muscles and tendons, such as the AT) by stroking the skin using an instrument while seated (in a chair) or kneeling on one knee (Figure 1b). The intervention was performed in the afternoon in the laboratory three times weekly (approximately 5 min each time) for 6 weeks. We supervised the intervention and ensured adherence to the protocol. In addition, during the intervention sessions, we frequently confirmed the absence of changes in the physical activities of the participants by asking them verbally.

The intervention included the following five components: (1) the entire posterior lower leg (approximately 60 s) (Figure 2a); (2) gastrocnemius and the area near the myotendinous junction between the gastrocnemius and AT (approximately 90 s) (Figure 2b); (3) SOL (approximately 45 s) (Figure 2c); (4) AT (approximately 60 s) (Figure 2d); and (5) posterior tibialis, flexor hallucis longus, and flexor digitorum longus (approximately 45 s). Based on a previous study [7], the strokes were performed from the long axis to the short axis along the pathway of the muscle or tendon fibers. Participants were instructed to create gentle and firm strokes that applied compression and shear stress to produce a pulling force in the engaged tissues around the edges of the instrument. They were also instructed to adjust the stimulation pressure and stroke rate accordingly to avoid excessive pain. In clinical practice, IASTM may be performed while holding the joint angle to stretch the muscles and tendons, allowing the deeper tissues to be accessed from the superficial layers. However, we cautiously avoided the effects of stretching on the soft tissues (the ankle joint was not dorsiflexed, and IASTM was performed from an anatomically upright position to a plantar flexion position). Before the start of the intervention period, participants received 60 min of training in IASTM by the protocol developer.

### 2.4. Measurement of the DFROM

An isokinetic dynamometer (BIODEX SYSTEM4, Biodex Medical Systems, Shirley, NY, USA) was used to measure the DFROM (Figure 3). The participant was positioned with the knee joint in full extension (hip joint at 60° flexion) and the ankle joint at 30° plantar flexion. The thigh and foot were fixed with belts to the seat and footplate of the isokinetic dynamometer, respectively. From that limb position, the ankle joint was dorsiflexed at approximately 2°/s to the angle at which the participant felt pain (4 on a visual analog scale of 1–5; 1: no pain at all, 2: mild stretch feeling, 3: moderate stretched feeling, 4: discomfort [defined as feeling pain], and 5: intolerable discomfort) [7]. The DFROM was defined as the movement of the ankle joint from 0° (anatomical position) to the maximum dorsiflexion angle during measurement. The torque around the plantar flexion direction at the maximum dorsiflexion angle was defined as the maximal passive torque. This measurement was performed five times, and the mean of three measurements represented the DFROM, excluding the largest and smallest values. The maximal passive torque was the average of the three values in the measurements used to calculate the DFROM. We instructed the participants to remain relaxed during the measurements and not resist passive dorsiflexion. Torque and ankle joint angle signals from the isokinetic dynamometer were converted at 1 kHz using an analog-to-digital (A/D) converter (PowerLab/16SP, ADInstruments, Sydney, Australia) and recorded on a personal computer using analysis software (LabChart8, v8.1.2version, ADInstruments). 

We recorded muscle activity in the lower leg muscles during DFROM measurements using surface electromyography (Wireless EMG system [Trigno], Delsys, Inc., Boston, MA, USA). We applied active surface electrodes (inter-electrode distance of 10 mm; Trigno Avanti Sensor, SP-W06-014 model, Delsys, Inc., Natick, MA, USA) to the muscle bellies of the MG, LG, SOL, and tibialis anterior (TA). The electrodes were placed at approximately 30% proximally along the length of the lower leg for the MG, LG, and TA and at approximately 30% distally along the length of the lower leg for the SOL. We shaved these sites and cleaned the skin using alcohol before application. EMG signals were also converted with an A/D converter and recorded using software (data smoothing on the software: band-pass filter, passband at 25–450 Hz, sampling frequency of 1 kHz) on a personal computer. Each muscle’s activity during the DFROM measurements was calculated as the root mean square (RMS), normalized by the RMS values during isometric plantar flexion and dorsiflexion at maximum voluntary contraction (MVC). Muscle activity during the DFROM measurements was comparable to that in a previous study [7] in all participants (mean RMS values of pre- and post-measurements for both conditions: MG—1.9 ± 1.1% MVCRMS; LG—1.8 ± 1.3% MVCRMS; SOL—2.7 ± 1.5% MVCRMS; and TA—1.0 ± 0.7% MVCRMS).

### 2.5. Measurement of Muscle and Tendon Stiffness

Shear wave elastography (Aixplorer, MSK mode, SuperSonic Imagine, Aix-en-Provence, France) was used to measure the MG and AT shear modulus as muscle and tendon stiffness indices, respectively. For the MG, the ankle joint angles during the shear modulus measurement were 0°, 10° dorsiflexion, and 20° dorsiflexion, referencing a previous study [15]. The participants’ posture during the shear modulus measurements was the same as during the DFROM measurement, with the participant seated with the knee in full extension on the isokinetic dynamometer (Figure 3). The MG measurement site was at the center of the muscle belly (approximately 30% proximally along the length of the lower leg). A linear ultrasound probe (bandwidth—2–10 MHz; scan width—50 mm; SL10-2, SuperSonic Imagine) was placed at the measurement site, and a long-axis B-mode ultrasound image (sampling frequency of 11 Hz) was captured with color mapping of the MG shear modulus value. We acquired five ultrasound images for each joint angle. Participants were instructed to relax during the measurements.

We used an elastography image analysis program (T.K.K. 5840, Takei Scientific Instruments, Niigata, Japan) to analyze the ultrasound images on a personal computer (Figure 4). We set a rectangular region of interest (ROI) for each image and obtained the mean value of the shear modulus within the ROI. The MG stiffness for each joint angle was defined as the mean of three out of the five shear modulus values obtained from the five ultrasound images, excluding the largest and smallest. 

The ankle joint angle for the AT shear modulus measurement was 20° plantar flexion. The participants’ posture for the measurements was the same as that for MG stiffness, except for the joint angle. Based on previous research, the AT measurement site was the free part of the tendon distal to the distal end of the SOL’s belly [16]. Ultrasound images were acquired and analyzed similarly to those for the MG to determine AT stiffness for each joint angle (Figure 4).

### 2.6. Statistical Analysis

All data are expressed as mean ± standard deviation. Two-way repeated measures analysis of variance [intervention (IASTM vs. control) × time (pre and post)] was performed for the dependent variables using statistical software (IBM SPSS statistics 24, SPSS Japan, Tokyo, Japan). If an interaction or a main effect of time was observed, a paired *t*-test was performed on each condition. A paired *t*-test was also used to ensure that there were no differences in the pre-intervention measurements between the conditions. Cohen’s *d* (post-hoc comparisons) and partial η^2^ (ηp^2^: ANOVA) were calculated for the effect size. Cohen’s *d* was obtained using the following equation: d = *M*_diff_/SD_pooled_ √2[1 − *r*], where *M*_diff_ is the difference between the mean value of the pre- and post-measurements, and *r* is the correlation between the mean values [17]. A priori statistical power analyses were used to estimate that this study design would require 14 participants for each of the 2 conditions (repeated measures ANOVA within factors; effect size, 0.4; power, 0.8; alpha level, 0.05) [18] using G*power 3. The effect size (Cohen’s *d*) was defined as |0.20| − |0.50| small effect, |0.50| − |0.80| medium effect, and >|0.80| large effect [18]. Statistical significance was set at a *p*-value of <0.05. 

## 3. Results

The DFROM showed a significant interaction between condition and time (ηp^2^ = 0.37; *p* = 0.016). Post hoc results showed that the DFROM improved significantly in the IASTM condition (22.6 ± 10.4° [Pre], 26.6 ± 11.7° [Post]; *p* = 0.024; d = 0.70), whereas the control condition did not change significantly (22.7 ± 9.7° [Pre], 23.0 ± 9.7° [Post]; *p* = 0.82; d = 0.072) (Figure 5a). A significant main effect of time was also observed for maximal passive torque (ηp^2^ = 0.30; *p* = 0.035). The maximal passive torque increased significantly under the IASTM condition (24.6 ± 11.5 Nm [Pre], 28.3 ± 14.7 Nm [Post]; *p* = 0.005; d = 1.38) and did not change significantly under the control condition (25.0 ± 15.0 Nm [Pre], 26.6 ± 12.9 Nm [Post]; *p* = 0.37; d = 0.26) (Figure 5b).

No interaction between condition and time or main effect of time was observed for MG stiffness at ankle joint angles of 0° and dorsiflexion of 10° (Table 1). A significant main effect of time was observed for MG stiffness at ankle dorsiflexion of 20° (ηp^2^ = 0.28; *p* = 0.042). However, post-hoc results showed no significant changes in the IASTM (*p* = 0.14) or control conditions (*p* = 0.086) (Table 1). In contrast, AT stiffness showed no interaction between condition and time or the main effect of time (Table 1). Cohen’s d values were as follows: IASTM group—MG stiffness of 0°, 0.51; dorsiflexion of 10°, 0.20; dorsiflexion of 20°, 0.42; AT stiffness, 0.013; control group—MG stiffness of 0°, 0.25; dorsiflexion of 10°, 0.23; dorsiflexion, 20°, 0.50; AT stiffness, 0.31.

## 4. Discussion

In the present study, a 6-week IASTM program improved DFROM and increased maximal passive torque during DFROM measurement. However, MG and AT stiffness did not change after the IASTM program.

The IASTM intervention was based on a protocol from a previous study [7] that observed an acute improvement in joint flexibility. Long-term implementation of the intervention may have contributed to the increased joint flexibility observed in the participants in this study. We set the period and frequency of the intervention to exceed the period and frequency used in flexibility (stretching) training, as recommended by the National Strength and Conditioning Association guidebook (≥5 weeks, ≥2 times weekly) [19]. Further studies are required to clarify the duration and frequency of IASTM required to improve flexibility. In sports, IASTM is often combined with another treatment modality (resistance exercise and/or stretching), suggesting that such mixed treatment modalities improve joint flexibility [8,9]. Notably, several weeks of continuous stretching (stretching training) effectively improve joint flexibility [11,20]. IASTM applies mechanical stimulation involving compression and shear stress to the muscle (overlying deep fascia) and tendon from the superficial layer (above the skin). On the other hand, stretching applies mechanical stimulation involving extension stress to the entire muscle–tendon tissue. IASTM and stretching use different approaches for muscle–tendon tissue, and greater benefits may be obtained when combined. In addition to stretching, the effectiveness of interventions that combine IASTM with various conditioning methods or resistance training should be verified in the future.

Maximal passive torque during DFROM measurements is used to indicate stretch tolerance [11,12]. Previous studies have suggested that modification of stretch tolerance is a factor contributing to improved joint flexibility with stretching training [11,12,20]. In the present study, the increase in maximal passive torque after the 6-week IASTM program implied a change in stretch tolerance. Therefore, our results suggest that a change in stretch tolerance contributes to improving joint flexibility through several weeks of continuous IASTM. It is believed that IASTM modulates proprioceptor responses in subcutaneous target muscles to reduce pain [13]. Likewise, in this study, IASTM possibly altered the response of proprioceptors in the target muscles under the skin, which increased maximal passive torque. Furthermore, a previous study that verified the acute effect reported that stretch tolerance did not change immediately after IASTM [7]. Therefore, the modulation of stretch tolerance may be induced by repeated and continuous IASTM within 6 weeks. The impact of continued IASTM on neurophysiological indices warrants further exploration.

In the present study, MG and AT stiffness did not change after the 6-week IASTM program. Musculotendinous stiffness affects the joint flexibility of the MTU and is altered by interventions such as stretching [11,20]. Our results suggest that continued IASTM does not affect musculotendinous stiffness. In previous studies, the training intensity and period can also affect the adaptation of muscle–tendon tissue [21,22,23]. Therefore, it is necessary to examine whether similar results would be obtained by changing elements of the IASTM intervention protocol (duration, frequency, and period, among others).

A key role of joint flexibility is its contribution to an athlete’s movement; an optimal level of flexibility is required for the movements performed in physical exercise and sports activities [19]. The results of this study suggest that IASTM improves joint flexibility in individuals without varied pathologies or injuries, showing that the 6-week IASTM program for the soft tissue of the posterior aspect of the lower leg improved DFROM by 18%. Previous studies of 6 weeks of static stretching (a classic flexibility improving exercise) in active young adults, such as the participants in our study, showed a 9–18% improvement in DFROM [24,25,26]. Therefore, IASTM is not inferior to static stretching in improving joint flexibility. IASTM and stretching are often combined in clinical settings. A greater effect than stretching alone or IASTM alone may be obtained in such cases. However, joint flexibility that is too high is believed to cause an increased risk of sports injuries [27]. Instructors should consider IASTM when planning their athletes’ training, in order to improve their joint flexibility. Additionally, stretching can alter the mechanical and neural properties of the MTU [10,11,12], whereas our results suggest that IASTM changes only the neural properties of the MTU. MTU stiffness should not be softened, depending on the intended sports event or athletic performance [28,29]. Therefore, IASTM is an option for athletes wishing to improve joint flexibility without altering the mechanical properties of the MTU.

This study has some limitations. First, the intensity of the IASTM treatment in the present study was determined subjectively by the participants, and the pressure or force of the instrument was not measured. Therefore, it is necessary to find an objectively effective stroke intensity using methods for assessing instrument pressure or force during IASTM. Second, our results were obtained from young and healthy participants. Therefore, it is unclear whether similar results would be obtained from participants with other characteristics (children, older people, or athletes). Third, in this study, one leg of each participant underwent a 6-week IASTM program on the posterior aspect of the lower leg, while the other leg did not undergo IASTM (control leg). Therefore, improving DFROM in one leg may impact the gait pattern in the other leg and affect muscle and tendon stiffness. However, it has been reported that the mean ankle joint range of motion during normal walking in healthy participants is 10.2° dorsiflexion and 14.2° plantarflexion, totaling approximately 25° [30]. In the present study, the participants’ mean DFROM before the intervention was well above the range of dorsiflexion used during normal walking (mean values before the intervention: IASTM condition—22.6°; control condition—22.7°). Therefore, it appears that the participants’ gait patterns did not change following the IASTM intervention, and this did not affect the results regarding muscle and tendon stiffness. Considering the application to sports, the participants performed the IASTM themselves in the present study. The intervention may have been more effective if a skilled therapist had performed the IASTM instead of the participants. 

Furthermore, our results were obtained from performing only IASTM. However, in routine practice, IASTM and other treatment modalities (stretching and resistance exercise, among others) are often combined in the sports field. Therefore, when implemented in routine practice, the effects on joint flexibility may be greater than those observed from IASTM alone. One advantage of IASTM is that the equipment is small and portable and can be used anywhere (at practice or competition venues). Based on our results, IASTM is a highly accessible flexibility training option for athletes who desire better joint flexibility.

## 5. Conclusions

The 6-week IASTM program for the posterior lower leg improved the DFROM by altering stretch tolerance without altering MG and AT stiffness. These results indicate that the IASTM program could improve joint flexibility.

## Figures and Tables

**Figure 1 sports-12-00150-f001:**
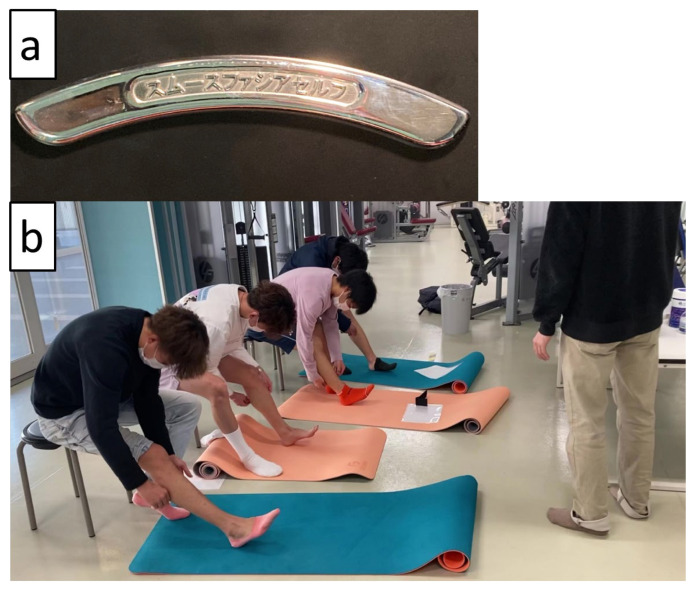
Photographs of the instruments used for instrument-assisted soft tissue mobilization (IASTM) (**a**) and the IASTM intervention (**b**) used in this study.

**Figure 2 sports-12-00150-f002:**
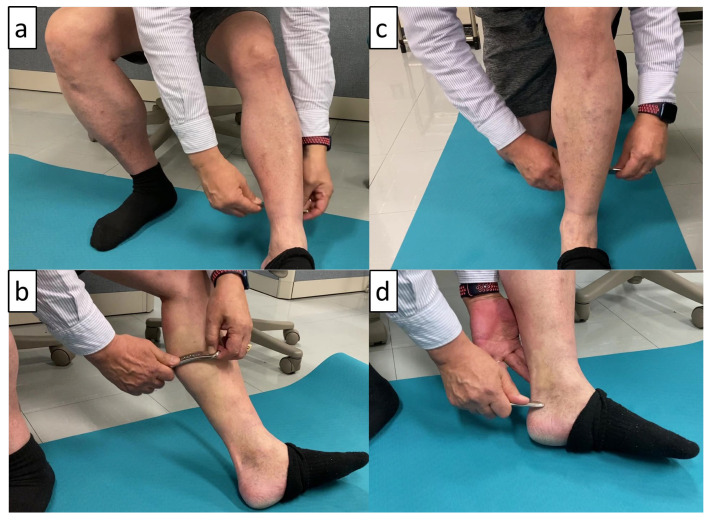
Instrument-assisted soft tissue mobilization (IASTM): (**a**) whole posterior lower leg; (**b**) gastrocnemii and part of muscle-tendon [gastrocnemii-Achilles tendon] junction; (**c**) soleus; and (**d**) Achilles tendon.

**Figure 3 sports-12-00150-f003:**
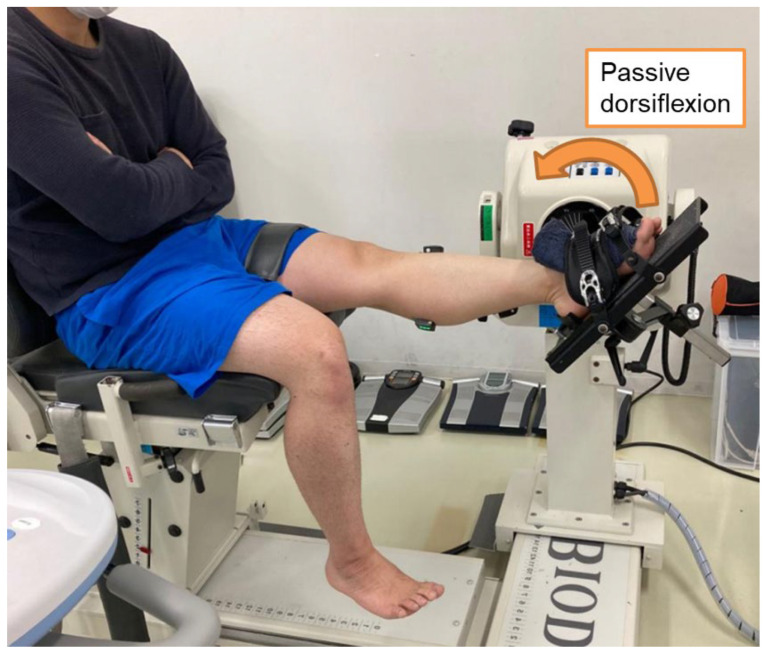
Illustration of the measurement of the ankle dorsiflexion range of motion.

**Figure 4 sports-12-00150-f004:**
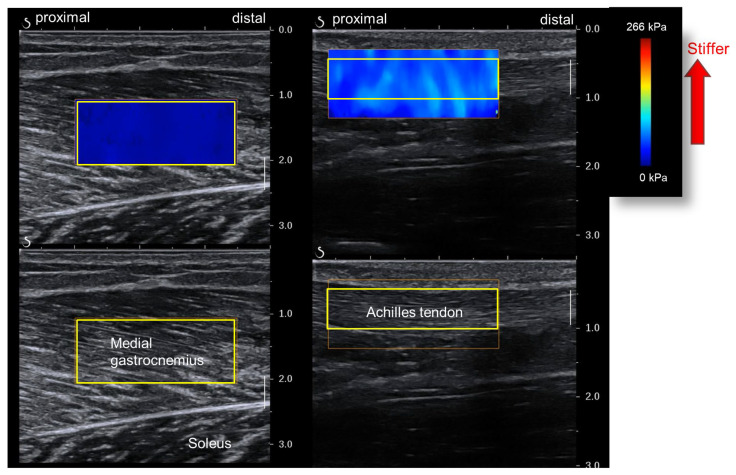
Analysis of muscle and Achilles tendon shear moduli wave velocities (yellow squares—regions of interest [ROIs]). The muscle shear modulus is obtained from the medial gastrocnemius (**left**). The Achilles tendon shear modulus is obtained from the free tendon part (**right**).

**Figure 5 sports-12-00150-f005:**
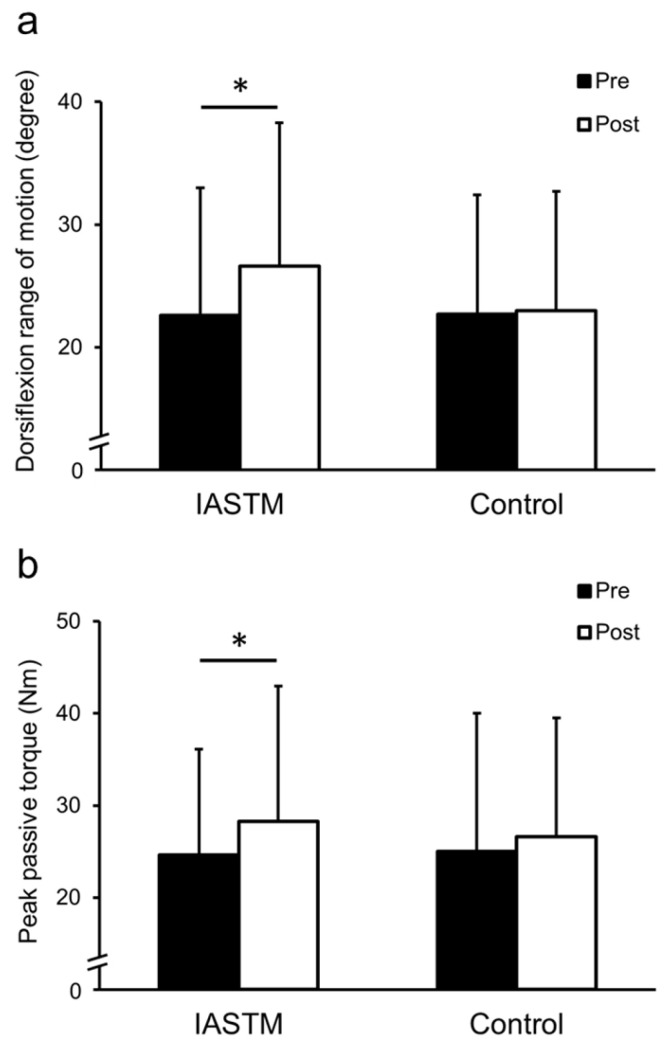
Dorsiflexion range of motion (**a**) and the peak passive torque (**b**) during dorsiflexion measurement in each condition. * Significantly changed compared with pre-intervention (*p* < 0.05). Values are expressed as mean ± standard deviation.

**Table 1 sports-12-00150-t001:** Changes in shear modulus values for the medial gastrocnemius and Achilles tendon stiffness in each condition.

	IASTM		Control	
	Pre	Post	Pre	Post
Medial gastrocnemius shear modulus (kPa)				
0° (neutral position)	10.3 ± 3.3	8.7 ± 1.5	10.6 ± 3.0	9.7 ± 3.0
DF 10°	16.7 ± 4.4	15.8 ± 2.9	17.4 ± 6.7	15.9 ± 4.8
DF 20°	32.3 ± 8.5	29.4 ± 7.1	33.7 ± 8.8	30.6 ± 8.9
Achilles tendon shear modulus (kPa)				
PF 20°	78.4 ± 24.1	78.8 ± 27.4	81.5 ± 17.7	86.6 ± 24.5

None: These parameters showed no significant changes relative to the pre-intervention value in any condition (*p* > 0.05). Values are expressed as mean ± standard deviation. IASTM—instrument-assisted soft tissue mobilization; DF—dorsiflexion; PF—plantar flexion.

## Data Availability

The data presented in this study are available on request from the corresponding author.

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
