# Peer review of "Effect of 6-Week Instrument-Assisted Soft Tissue Mobilization on Joint Flexibility and Musculotendinous Properties"

_sports, 2024, doi:10.3390/sports12060150_

Round 1

Reviewer 1 Report

Comments and Suggestions for Authors

Dear authors,

This is an interesting work. Please see my comments below.

Overall comment:

The discussion requires a bit of more support from previous work regarding the effect of the different characteristics of mechanical loading (stretching induces mechanical tension on muscle and tendinous tissue and so the IASTM modality by applying pressure along the muscle fiber's orientation) on the muscle-tendon unit, especially on the tendon since its properties (mostly the tendon's material properties) respond differently compared to the muscular response.

Abstract = no comments here

Introduction

Lines 52-53: I disagree with defining IASTM exercise, because it is not. It is a method assisting the ligaments gain a greater range of motion similar to the foam rolling method. One would use it either before or after exercise but not during an exercise program or a training. Therefore, I suggest wording it as a "treatment modality".

Methods

Line 82: please provide specific details about the intensity and the exercise routine (type of activities) of participants. 

Lines 114-115: were did the participants perform the protocol? at home or at the Lab? did the authors ensure that participants perform the intervention protocol at the same period within their day, for example in the morning or the afternoon? Did the authors ask the participants to keep a log book of their adherence with the intervention protocol? For example, 

results could be affected if subjects did not do the intervention around the same time of day. An increase in tissue stiffness (lumbodorsal fascia in mice) has been shown after static stretching that was based on a higher matrix hydration compared to initial levels, which was characterized as a super-compensation phenomenon. 

Schleip R, Duerselen L, Vleeming A, Naylor IL, Lehmann-Horn F, Zorn A, Jaeger H, Klingler W. Strain hardening of fascia: static stretching of dense fibrous connective tissues can induce a temporary stiffness increase accompanied by enhanced matrix hydration. J Bodyw Mov Ther. 2012 Jan;16(1):94-100. doi: 10.1016/j.jbmt.2011.09.003

Line 170: please specify the sampling frequency of EMG data.

Lines 187-189: same here,what was the sampling frequency of U/S?

Line 196: correct to "as the mean of the three out of five..."

Line 211: the IBM SPSS software should be reported on its own outside the parentheses. In the parentheses, typically the factors and their levels are reported, so it is suggested to write "A two-way repeated measures analysis of variance (intervention (IASTM vs control) x time (Pre vs Post)) was performed in the dependent variables".

Line 213-214: Why not use pairwise comparisons with Bonferroni correction? This is typical for 2way RM anova analysis.

Section Results: please report the Cohen's d values for all the results, regardless of statistical significance.

Discussion

Line 257: the authors write "...which may have influenced the results." PLease be more clear about this argument.

Lines 265-266:  "IASTM and stretching use different approaches for muscle-tendon tissue,..." agian it is not clear what the authors want to say. Please specify.

Line 276: "additionally" can be replaced by similarly or likewise.

Lines 286-287: references are needed to support this argument "However, the training volume and period can also affect the adaptation of muscle-tendon tissue". the authors could check the references by the Berlin group, for example 

Arampatzis, A., Karamanidis, K., & Albracht, K. (2007). Adaptational responses of the human Achilles tendon by modulation of the applied cyclic strain magnitude. The Journal of Experimental Biology, 210, 2743–2753. https://doi.org/10.1242/jeb.003814

Arampatzis A, Peper, A., Bierbaum, S., & Albracht, K. (2010). Plasticity of human Achilles tendon mechanical and morphological properties in response to cyclic strain. Journal of Biomechanics, 43(16), 3073–3079. https://doi.org/10.1016/j.jbiomech.2010.08.014

Arampatzis A, Mersmann, F., & Bohm, S. (2020). Individualized muscle-tendon assessment and training. Frontiers in Physiology, 11, 723. https://doi.org/10.3389/fphys.2020.00723

Bohm, S., Mersmann, F., Tettke, M., Kraft, M., & Arampatzis, A. (2014). Human Achilles tendon plasticity in response to cyclic strain: effect of rate and duration. The Journal of Experimental Biology, 217, 4010–4017. https://doi.org/10.1242/jeb.112268

Bohm, S., Mersmann, F., & Arampatzis, A. (2015). Human tendon adaptation in response to mechanical loading: a systematic review and meta-analysis of exercise intervention studies on healthy adults. Sports Medicine - Open, 1(1), 7. https://doi.org/10.1186/s40798-015-0009-9

Comments on the Quality of English Language

No comments here.

Author Response

Response to Reviewer 1:

Introduction

Lines 52-53: I disagree with defining IASTM exercise, because it is not. It is a method assisting the ligaments gain a greater range of motion similar to the foam rolling method. One would use it either before or after exercise but not during an exercise program or a training. Therefore, I suggest wording it as a "treatment modality".

Response: Thank you for your careful review and valuable comments and suggestions. We have revised the sentence and used the term “treatment modality” (lines 52-53).

Methods

Line 82: please provide specific details about the intensity and the exercise routine (type of activities) of participants.

Response: Thank you for your helpful comment. We have added information regarding the intensity and exercise routine (type of activities) of participants (lines 81-84).

Lines 114-115: were did the participants perform the protocol? at home or at the Lab? did the authors ensure that participants perform the intervention protocol at the same period within their day, for example in the morning or the afternoon? Did the authors ask the participants to keep a log book of their adherence with the intervention protocol? For example,

results could be affected if subjects did not do the intervention around the same time of day. An increase in tissue stiffness (lumbodorsal fascia in mice) has been shown after static stretching that was based on a higher matrix hydration compared to initial levels, which was characterized as a super-compensation phenomenon.

Schleip R, Duerselen L, Vleeming A, Naylor IL, Lehmann-Horn F, Zorn A, Jaeger H, Klingler W. Strain hardening of fascia: static stretching of dense fibrous connective tissues can induce a temporary stiffness increase accompanied by enhanced matrix hydration. J Bodyw Mov Ther. 2012 Jan;16(1):94-100. doi: 10.1016/j.jbmt.2011.09.003

Response: Thank you for your valuable insights and suggestions. We have added the relevant information in the revised manuscript (lines 119-123). Participants performed IASTM in the laboratory in the afternoon.

Line 170: please specify the sampling frequency of EMG data.

Lines 187-189: same here,what was the sampling frequency of U/S?

Line 196: correct to "as the mean of the three out of five..."

Line 211: the IBM SPSS software should be reported on its own outside the parentheses. In the parentheses, typically the factors and their levels are reported, so it is suggested to write "A two-way repeated measures analysis of variance (intervention (IASTM vs control) x time (Pre vs Post)) was performed in the dependent variables".

Response: Thank you for your valuable suggestions. We have added all the relevant information in the revised manuscript (lines 179-180; line 198-199; lines 205-206; lines 220-223).

Line 213-214: Why not use pairwise comparisons with Bonferroni correction? This is typical for 2way RM anova analysis.

Response: Thank you for your advice. We agree with you; however, we believe that our methodology is also not incorrect. For two-group comparisons, post-hoc testing using t-tests has also been used in previous studies.

Ikeda, N.; Otsuka, S.; Kawanishi, Y.; Kawakami, Y. Effects of instrument-assisted soft tissue mobilization on musculoskeletal properties. Med Sci Sports Exerc 2019, 51, 2166–2172. DOI:10.1249/MSS.0000000000002035.

Ikeda, N.; Yonezu, T.; Kawakami, Y. Minute oscillation stretching: A novel modality for reducing musculo-tendinous stiffness and maintaining muscle strength. Scand J Med Sci Sports 2021, 31, 104–114. DOI:10.1111/sms.13830.

Section Results: please report the Cohen's d values for all the results, regardless of statistical significance.

Response: Thank you for pointing this out. We have added Cohen’s d values for all the results (line 238; line 243; lines 254-257).

Discussion

Line 257: the authors write "...which may have influenced the results." PLease be more clear about this argument.

Lines 265-266:  "IASTM and stretching use different approaches for muscle-tendon tissue,..." agian it is not clear what the authors want to say. Please specify.

Response: We apologize for the lack of clarity. We have revised the sentences for better clarity (lines 269–271; lines 278–281).

Line 276: "additionally" can be replaced by similarly or likewise.

Response: Thank you for your advice. We have replaced “similarly” with “likewise” (line 293).

Lines 286-287: references are needed to support this argument "However, the training volume and period can also affect the adaptation of muscle-tendon tissue". the authors could check the references by the Berlin group, for example

Arampatzis, A., Karamanidis, K., & Albracht, K. (2007). Adaptational responses of the human Achilles tendon by modulation of the applied cyclic strain magnitude. The Journal of Experimental Biology, 210, 2743–2753. https://doi.org/10.1242/jeb.003814

Arampatzis A, Peper, A., Bierbaum, S., & Albracht, K. (2010). Plasticity of human Achilles tendon mechanical and morphological properties in response to cyclic strain. Journal of Biomechanics, 43(16), 3073–3079. https://doi.org/10.1016/j.jbiomech.2010.08.014

Arampatzis A, Mersmann, F., & Bohm, S. (2020). Individualized muscle-tendon assessment and training. Frontiers in Physiology, 11, 723. https://doi.org/10.3389/fphys.2020.00723

Bohm, S., Mersmann, F., Tettke, M., Kraft, M., & Arampatzis, A. (2014). Human Achilles tendon plasticity in response to cyclic strain: effect of rate and duration. The Journal of Experimental Biology, 217, 4010–4017. https://doi.org/10.1242/jeb.112268

Bohm, S., Mersmann, F., & Arampatzis, A. (2015). Human tendon adaptation in response to mechanical loading: a systematic review and meta-analysis of exercise intervention studies on healthy adults. Sports Medicine - Open, 1(1), 7. https://doi.org/10.1186/s40798-015-0009-9

Response: Thank you for the excellent recommendations. We have revised the sentence and added supporting references (lines 303-304).

Reviewer 2 Report

Comments and Suggestions for Authors

Review on the article: Effect of 6-week instrument-assisted soft tissue mobilization on joint flexibility and musculoskeletal properties

Title: I would suggest correcting the title of the article since "musculoskeletal properties" is a broad term and this article specifically focuses on elasticity (maybe musculotendinous properties).

Abstract: Authors state that "However, no studies have verified whether several weeks of continuous IASTM alone can alter stretch tolerance in healthy participants". My question is: how many articles state what there is a need to increase stretch tolerance in healthy people? What is the reason to do this? Who determined the duration of the intervention? Why 6 weeks, not 4 or 8?

"No measurement changed significantly in the control group - this sentence is unclear.

Introduction: In line 30 the instrument is called spurtle-like in line 10 - spatula-like. What is the correct name of instrument?

For me in unclear how IASTM is defined. In one place (lines 39, 42, 52, 60) it is exercise, then it is a protocol (line 103), and then a program (line 73, 95, ...), intervention (line 66).

Please justify sentence: "IASTM alone as an exercise for improving joint flexibility is vital".

Methods: In participants inclusion criteria is not included varicose vein, but in Figure 2. especially a and c, seems that participant has vascular problems. Might is be in participant aged 21?

In line 97 authors write that "Both legs were measured in all participants". This statement needs clarification. Do you mean that all measurements were taken in the same order and on both legs. Whether it has been determined which leg is dominant. How this might have affected the results ?

I doubt whether this intervention can be called a protocol, maybe the word "intervention" fits into context better. I would suggest using: "The intervention was carried out according to the protocol, developed by..."

Was the intervention supervised by the same protocol developer?

Authors write that "This protocol was based on findings regarding acute improvements in joint flexibility". What is acute improvement? Immediately after intervention? After 6 weeks? If after 6 weeks it is not acute.

Authors write: "Muscle activity during the DFROM measurements was low in all participants (mean RMS values of pre- and post-measurements for both conditions: MG: 1.9 ± 1.1% MVCRMS, LG: 1.8 ± 1.3% MVCRMS, SOL: 2.7 ± 1.5% MVCRMS, and TA: 1.0 ± 0.7% MVCRMS)".This sentence is unclear. Is it normal activity, or low, or unexpected?

Discussion. It is unclear what authors wanted to say: "IASTM protocol was based on a protocol from a previous study [7] that observed an acute improvement in joint flexibility, which may have influenced the results". What influenced the results: acute improvement ar the same protocol? Please reword this sentence and explain properly.

Seems that authors are contradicting themselves. The previously developed protocol is ineffective and they still do research on it (lines 256-260).

Discussion section is unclear. Authors discuss interventions what were not applied in the present study "Our results suggest that continued IASTM does not affect musculotendinous stiffness. However, the training volume and period can also affect the adaptation of muscle-tendon tissue". Where from this training volume comes? How it is related to research done?

In line 292 authors write: "They showed that the 6-week..." Who are they?

The conclusion cannot be generalized, and authors cannot claim that the intervention used is effective in improving joint flexibility, as it was only applied to one segment of the body. There should be a balance between mobility and stability.

Authors in the article use "We used, we placed...(especially between lines 162-172), Our...". Better to avoid these expressions.

Comments on the Quality of English Language

I would suggest proofreading and editing English to eliminate ambiguities. 

Author Response

Response to Reviewer 2:

Title: I would suggest correcting the title of the article since "musculoskeletal properties" is a broad term and this article specifically focuses on elasticity (maybe musculotendinous properties).

Response: Thank you for your careful review and valuable comments and suggestions. We have replaced “musculoskeletal properties” with “musculotendinous properties” in the title and revised manuscript (line 3; line 13; line 37; line 98).

Abstract: Authors state that "However, no studies have verified whether several weeks of continuous IASTM alone can alter stretch tolerance in healthy participants". My question is: how many articles state what there is a need to increase stretch tolerance in healthy people?

"No measurement changed significantly in the control group - this sentence is unclear.

Response: Thank you for your helpful comments. We apologize for the mistake. We have replaced “stretch tolerance” with “joint flexibility and the musculotendinous properties” (line 11-12). We have modified the Abstract according to your suggestions (lines 20).

What is the reason to do this? Who determined the duration of the intervention? Why 6 weeks, not 4 or 8?

Response: Thank you for your question. We determined the intervention period according to the recommendations for flexibility training by the National Strength and Conditioning Association guidebook (≥5 weeks). In addition, the intervention period was set at 6 weeks because an 8-week period would have resulted in excessive strain on the participants; moreover, it enabled us to perform comparisons with the results of previous studies on 6-week stretching training (see Discussion section).

Introduction: In line 30 the instrument is called spurtle-like in line 10 - spatula-like. What is the correct name of instrument?

Response: We apologize for the discrepancy. We have revised the term to “spurtle-like” in line 10.

For me in unclear how IASTM is defined. In one place (lines 39, 42, 52, 60) it is exercise, then it is a protocol (line 103), and then a program (line 73, 95, ...), intervention (line 66).

Please justify sentence: "IASTM alone as an exercise for improving joint flexibility is vital".

Response: We apologize for the inconsistency. We have revised these sentences (line 39, line 41, line 43, among others). We have revised “exercise” to “treatment modality” throughout the manuscript.

Methods: In participants inclusion criteria is not included varicose vein, but in Figure 2. especially a and c, seems that participant has vascular problems. Might is be in participant aged 21?

Response: Thank you for pointing this out. The Figure is a representative figure for demonstration by the protocol developer.

In line 97 authors write that "Both legs were measured in all participants". This statement needs clarification. Do you mean that all measurements were taken in the same order and on both legs. Whether it has been determined which leg is dominant. How this might have affected the results ?

Response: We apologize for the lack of clarity. We have added the required information in the “Study design” section (lines 101-103). The right leg was the dominant leg in all participants. The leg on which IASTM, regardless of being dominant or non-dominant, was randomized. This has been mentioned in the revised manuscript (lines 102-103). We believe this did not affect the results of the present study.

I doubt whether this intervention can be called a protocol, maybe the word "intervention" fits into context better. I would suggest using: "The intervention was carried out according to the protocol, developed by..."

Response: Thank you for your suggestion. We have incorporated these changes in the revised manuscript (lines 108-112). We have also revised “protocol” to “intervention” at other relevant instances in the manuscript.

Was the intervention supervised by the same protocol developer?

Response: Thank you for your question. The intervention was supervised by us with guidance from the protocol developer. Participants received 60 minutes of direct instruction from the protocol developer before the start of the intervention period. We have clarified this in the revised manuscript (line 121; lines 142-143).

Authors write that "This protocol was based on findings regarding acute improvements in joint flexibility". What is acute improvement? Immediately after intervention? After 6 weeks? If after 6 weeks it is not acute.

Authors write: "Muscle activity during the DFROM measurements was low in all participants (mean RMS values of pre- and post-measurements for both conditions: MG: 1.9 ± 1.1% MVCRMS, LG: 1.8 ± 1.3% MVCRMS, SOL: 2.7 ± 1.5% MVCRMS, and TA: 1.0 ± 0.7% MVCRMS)".This sentence is unclear. Is it normal activity, or low, or unexpected?

Response: Thank you for pointing this out. We have rewritten these sentences for better clarity (lines 113-114; lines 182-184).

Discussion. It is unclear what authors wanted to say: "IASTM protocol was based on a protocol from a previous study [7] that observed an acute improvement in joint flexibility, which may have influenced the results". What influenced the results: acute improvement ar the same protocol? Please reword this sentence and explain properly.

Response: Thank you for your advice. We have revised this sentence (lines 269-271).

Seems that authors are contradicting themselves. The previously developed protocol is ineffective and they still do research on it (lines 256-260).

Response: We apologize for the inconsistency. We have removed the text (lines 271, the following part of “…in the results of this study.”).

Discussion section is unclear. Authors discuss interventions what were not applied in the present study "Our results suggest that continued IASTM does not affect musculotendinous stiffness. However, the training volume and period can also affect the adaptation of muscle-tendon tissue". Where from this training volume comes? How it is related to research done?

Response: Thank you for your valuable comments. We have revised these sentences for better clarity (lines 303-306).

In line 292 authors write: "They showed that the 6-week..." Who are they?

Response: We apologize for the lack of clarity. We were referring to the results of our study. We have revised and connected the sentences to indicate this (lines 309-311).

The conclusion cannot be generalized, and authors cannot claim that the intervention used is effective in improving joint flexibility, as it was only applied to one segment of the body. There should be a balance between mobility and stability.

Response: We agree with your comments. We have revised the concluding sentence accordingly (lines 344-346).

Authors in the article use "We used, we placed...(especially between lines 162-172), Our...". Better to avoid these expressions.

Response: Thank you for your advice. We have attempted to use the passive voice as much as possible.

Reviewer 3 Report

Comments and Suggestions for Authors

This paper is a repetition of the protocol the authors have previously used in 2019 but in the current manuscript, they extended the treatment to 6 weeks. 

The main flaws of this paper are 1) the lack of control in the administration of the treatment since it is self-administered compared to their 2019 paper and 2) a doubt about their sample size calculation with a effect size chosen to be too quite small (0,4).  I tried to reproduce their sample size calculation on G*Power without success. Finally, 3) inhibition to pain sensation with the used of the device during treatment may be associated to the improvement of the range of motion observed.    

Authors should justify the need to improve muscle and joint flexibility in the lower limb and during sport's participation.

Authors should also discuss about the control of activities within the 6-week protocol to ensure that subjects did not modify their physical activities.  Authors also have to convince me that improving range of motion in one leg is not modifying the gait pattern of the other limb and neutralized the effect of muscle and tendon stiffness.

Author Response

Response to Reviewer 3:

The main flaws of this paper are 1) the lack of control in the administration of the treatment since it is self-administered compared to their 2019 paper and 2) a doubt about their sample size calculation with a effect size chosen to be too quite small (0,4).  I tried to reproduce their sample size calculation on G*Power without success. Finally, 3) inhibition to pain sensation with the used of the device during treatment may be associated to the improvement of the range of motion observed.   

 Response: Thank you for your comments.

1) We have added information regarding control in the administration of the treatment (lines 119-123).

2) In case of ANOVA, effect sizes (f2) of 0.1, 0.25, and 0.4 are termed small, medium, and large, respectively, unlike those for Cohen’s d (Cohen 1988).

3) We agree with your comments. Stretch tolerance is an indicator of pain sensation, and we suggest that its increase contributed to an improvement in ROM. We have edited the text to improve clarity regarding stretch tolerance (lines 16-17; lines 344-346).

Cohen, J. Statistical Power Analysis for the Behavioral Sciences. 2nd ed.; Lawrence Erlbaum Associates: Hillsdale, USA, 1988.

Authors should justify the need to improve muscle and joint flexibility in the lower limb and during sport's participation.

Response: Thank you for your advice. We have added the relevant sentences (lines 307-309) in the revised manuscript. In this study, the triceps surae was targeted because it is easy to evaluate as a skeletal muscle-tendon model.

Authors should also discuss about the control of activities within the 6-week protocol to ensure that subjects did not modify their physical activities. 

Response: Thank you for pointing this out. We have added the relevant information in the revised manuscript (lines 121-123).

Authors also have to convince me that improving range of motion in one leg is not modifying the gait pattern of the other limb and neutralized the effect of muscle and tendon stiffness.

Response: Thank you for your valuable insights. It has been reported that the mean ankle joint range of motion during normal walking in healthy subjects is 10.2° dorsiflexion and 14.2° plantarflexion (total of approximately 25°) (Nordin & Frankel 2012)). In the present study, the participants’ mean ankle dorsiflexion range of motion before the intervention was well above the range of dorsiflexion used during normal walking (mean values before the intervention: IASTM group, 22.6°; control group, 22.7°). Therefore, we speculate that the gait pattern of the participants did not change following the IASTM intervention and did not affect the results of muscle and tendon stiffness.

Nordin, M. & Frankel, V.H. Basic Biomechanics of the Musculoskeletal System. 4th ed,; Wolters Kluwer Health: Philadelphia, USA, 2012.

Round 2

Reviewer 2 Report

Comments and Suggestions for Authors

The article is improved. 

No more comments from my side. 

Author Response

Thank you for your valuable time.

Reviewer 3 Report

Comments and Suggestions for Authors

Dear authors,

I am surprised that in the new version of your manuscript, you didn’t address how improving range of motion in one leg is not modifying the gait pattern of the other limb and neutralized the effect of muscle and tendon stiffness. A control group without intervention is the only way to make this study, a sound scientific study.

I was even more surprised to find that you choose to significantly alter your conclusion “These results suggest that the IASTM program improves joint flexibility by changing neural properties such as the pain sensation without affecting the mechanical properties of the treated muscles” compared with their previous conclusion “This indicates that an IASTM program could effectively improve joint flexibility”.

I do not support this new conclusion in any way and I find it inappropriate to make such changes in your manuscript during a second review process.

Author Response

Responses to Reviewer 3 Comments:

I am surprised that in the new version of your manuscript, you didn’t address how improving range of motion in one leg is not modifying the gait pattern of the other limb and neutralized the effect of muscle and tendon stiffness. A control group without intervention is the only way to make this study, a sound scientific study.

Response: Thank you for your valuable comment. As per your suggestion, we have modified the limitations portion of the manuscript as follows:

“Third, in this study, one leg of each participant underwent a 6-week IASTM program on the posterior aspect of the lower leg, while the other leg did not undergo IASTM (control leg). Therefore, improving DFROM in one leg may impact the gait pattern in the other leg and affect muscle and tendon stiffness. However, it has been reported that the mean ankle joint range of motion during normal walking in healthy participants is 10.2° dorsiflexion and 14.2° plantarflexion, totaling approximately 25° [30]. In the present study, the participants’ mean DFROM before the intervention was well above the range of dorsiflexion used during normal walking (mean values before the intervention: IASTM condition, 22.6°; control condition, 22.7°). Therefore, it appears that the participants' gait patterns did not change following the IASTM intervention, and this did not affect the results regarding muscle and tendon stiffness.” (lines 330–341)

I was even more surprised to find that you choose to significantly alter your conclusion “These results suggest that the IASTM program improves joint flexibility by changing neural properties such as the pain sensation without affecting the mechanical properties of the treated muscles” compared with their previous conclusion “This indicates that an IASTM program could effectively improve joint flexibility”.

I do not support this new conclusion in any way and I find it inappropriate to make such changes in your manuscript during a second review process.

Response: Thank you for your valuable comment. As per your suggestion, we have modified the original sentence as follows:

“These results indicate that the IASTM program could improve joint flexibility.” (lines 354–355).

Please note that we originally removed the word “effective” based on a following comment made by the other reviewer during the first round of editing.

Reviewer’s comment: “The conclusion cannot be generalized, and authors cannot claim that the intervention used is effective in improving joint flexibility, as it was only applied to one segment of the body. There should be a balance between mobility and stability.”